# Comparing the effectiveness and cost-effectiveness of self-management interventions in four high-priority chronic conditions in Europe (COMPAR-EU): a research protocol

Marta Ballester [1,2,3] Carola Orrego,[1,2,3] Monique Heijmans,[4]
Pablo Alonso-Coello,[5,6] Matthijs Michaël Versteegh,[7] Dimitri Mavridis,[8,9]
O Groene,[10] Kaisa Immonen,[11] Cordula Wagner,[4] Carlos Canelo-Aybar,[5,6]
Rosa Sunol,[1,2,3] for the COMPAR-EU consortium

For numbered affiliations see end of article.

**Correspondence to**
Marta Ballester;
mballester@fadq.org

## ABSTRACT

**Introduction** Population ageing and increasing chronic illness burden have sparked interest in innovative care models. While self-management interventions (SMIs) are drawing increasing attention, evidence of their efficacy is mostly based on pairwise meta-analysis, generally derived from randomised controlled trials comparing interventions versus a control or no intervention. As such, relevant efficacy data for comparisons among different SMIs that can be applied to specific chronic conditions are missing. Therefore, the relevance of the available evidence for decision-making at clinical, organisational and policy levels is limited.

**Aim** To identify, compare and rank the most effective and cost-effective SMIs for adults with four high-priority chronic conditions: type 2 diabetes, obesity, chronic obstructive pulmonary disease, and heart failure.

**Methods and analysis** All activities will be conducted as part of the cost-effectiveness of self-management interventions in four high-priority chronic conditions in Europe(COMPAR-EU, Comparing effectiveness of self-management interventions in 4 high priority chronic diseases inEurope) Project, an European Union (EU)-funded project designed to bridge the gap between current knowledge and practice on SMIs. In the first phase of the project, we will develop and validate a taxonomy, and a Core Outcome Set for each condition. These activities will inform a series of systematic review and network meta-analysis about the effectiveness of SMIs. We will also perform a cost-effectiveness analysis of the most effective SMIs and an evaluation of contextual factors. We will finally develop tailored decision-making tools for the different relevant stakeholders.

**Ethics and dissemination** Ethical approval was obtained from the local ethics committee (University Institute for Primary Care Research - IDIAP Jordi Gol). All patients and other stakeholders will provide informed consent prior to participation. This project has been funded by the EU Horizon 2020 research and innovation programme (grant agreement no. 754936). Results will be of interest to relevant stakeholder groups (patients, professionals, managers, policymakers and industry), and will be disseminated in a tailored multi-pronged approach that will include deployment of an interactive platform.

### Strengths and limitations of this study

► The project will result in the largest network meta-analysis (NMA) of complex self-management interventions (SMIs).
► SMIs are inconsistently defined across the literature potentially generating a high level of heterogeneity for the NMA, which we will mitigate by developing a validated taxonomy.
► The development of Core Outcome Sets with input from patients and other stakeholders for each chronic condition will ensure that outcomes assessed in the NMA are relevant to the target users.
► The comparative effectiveness analysis via NMA, cost-effectiveness and contextual factors evaluation will provide new knowledge that should facilitate future implementation of successful SMIs.
► An interactive platform will facilitate access to decision-making tools relevant to the specific needs of the different target users.

## INTRODUCTION

As population ageing accelerates worldwide, chronic illness will place an increasing burden on society and healthcare systems.[1] Chronic conditions affect over 80% of people aged over 65 in Europe and account for an estimated 77% of disease burden, as measured by disability-adjusted life years.[2] Furthermore, between 70% and 80% of healthcare costs in Europe can be attributed to chronic disease, and the current €700 billion expenditure is expected to rise.[3]

Self-management support has become a key strategy for addressing chronic disease burden,[4] contributing to the paradigm shift from a paternalistic model where patients are viewed as passive recipients of care, towards more equitable and collaborative models of clinician–patient interaction.[5]

Cost-effectiveness of self-management interventions in four high-priority chronic conditions in Europe (COMPAR-EU) is an EU-funded project designed to bridge the gap between current knowledge and practice on self-management interventions (SMIs). For the purpose of this project, we define self-management as 'actions that individuals, families, and communities engage in to promote, maintain, or restore health and cope with illness and disability, with or without the support of health professionals, and including but not limited to self-prevention, self-diagnosis, self-medication, and coping with illness and disability'.[6]

Self-management of a chronic condition requires self-efficacy, largely understood as a person's confidence in their ability to cope with their illness.[7] To be self-efficacious, people need special skills to cope with the consequences of the disease, including monitoring symptoms and clinical markers, understanding the implications of these, and adjusting behaviours treatment accordingly.

SMIs are supportive interventions systematically delivered or led by healthcare staff or other patients with the aim of building patients' confidence and equipping them with the necessary skills. Their purpose is to actively engage patients (and informal caregivers where appropriate) in the management of their disease.[8] As such, they are more than merely didactic, instructional programmes, as their primary objective is to bring about changes in behaviour and trigger a sequence of positive knock-on effects.[9]

SMIs are complex interventions,[9] typically characterised by multiple factors (components, formats, settings, target behaviours, etc) that interact over time as participants move back and forth between the intervention processes and everyday life, which can entail challenges in measuring effectiveness. Despite this, there is promising evidence that SMIs, under given conditions, can improve clinical outcomes in numerous chronic conditions, such as diabetes (reduction of haemoglobin A1c (HbA1C) levels),[10] obesity (reduction of weight loss),[11 12] chronic obstructive pulmonary disease (COPD) (improvement of dyspnoea)[13] and heart failure (reduction of mortality).[14] SMIs have also been associated with improvements in patient-reported outcomes, such as quality of life, and more specific disease measures,[15] such as self-efficacy[16] and adherence.[17]

Evidence on the efficacy of SMIs to date has mostly come from systematic reviews that have employed pairwise meta-analysis of randomised controlled trials (RCTs). Systematic reviews pool evidence from RCTs comparing the same interventions and have long been considered as the highest standard in evidence-based healthcare. Pairwise meta-analysis, however, leaves a crucial gap, because it requires the RCT's that are pooled to have included the same interventions. To provide decision-makers, clinicians and patients with solid evidence on how effective an SMI is for a given outcome and disease, multiple interventions need to be compared. In COMPAR-EU, we plan to do this using NMA to assess the relative effectiveness of SMIs in four chronic conditions.

NMA synthesises direct and indirect evidence across a network of multiple interventions. This method has numerous advantages: it provides more precise effect estimates, allows for the estimation of relative effectiveness between interventions that have not been compared directly and provides a ranking of interventions by effectiveness, presenting thus a potential analytical advantage.[18–22] NMA has been in use for some years, an empirical evaluation of 456 NMAs published up to 2015 showed that just 16% of these addressed complex interventions[23] such as SMIs.[24]

To support future reimbursement decisions based on estimates of the long-term effects of SMIs, we will develop simulation models in the four disease areas which will translate benefits as identified by the NMA into long-term health benefits expressed in quality-adjusted life years (QALYs), which enables us to compare not only SMIs within but also across the four conditions. The simulation models will also include a cost-effectiveness component. Cost-effectiveness analyses are important as they help to prioritise health expenditure. There is evidence that certain SMIs are cost-effective. Weight reduction in obesity, for example, can produce short-term savings and increase the chances of remaining in employment,[25] resulting in additional societal gains, while COPD management interventions can improve quality of life at generally acceptable societal costs,[26] and in some cases even result in short-term healthcare cost savings.[27] Secondary cardiovascular risk prevention programmes have the potential to reduce direct healthcare expenditure and improve health outcomes.[28] While these results have not yet been structured into a consolidated body of knowledge, they do indicate that investing in SMIs may be cost-effective.

A better understanding of facilitators and barriers to successful programme implementation is also essential. Contextual factors at various levels (patient, patient–provider interaction, organisation and system) can all influence SMI uptake, engagement and success. Improving our understanding of how these factors influence the effectiveness and cost-effectiveness of complex SMIs for chronic illness is much needed.

## Aim

COMPAR-EU is a multimethod interdisciplinary project that will run from 2018 to 2022. The project has been designed to help bridge the gap between current knowledge and practice in SMIs for chronic illness. Its aim is to identify, compare and rank the most effective SMIs for adults living with four high-priority chronic conditions (type 2 diabetes, obesity, COPD and heart failure), and

among these interventions, categorise the most cost-effective and feasible SMIs. The results of the project will facilitate informed decision-making and support the implementation of best practices in different health-care contexts through an interactive platform featuring decision-making tools, and other end products for poli-cymakers, guideline developers, researchers, healthcare professionals, patients and industry.

The specific objectives are to (1) validate a taxonomy of SMIs; (2) identify and prioritise SMI outcomes from the perspective of both patients and practitioners, culmi-nating in a Core Outcome Set (COS) for each condition; (3) carrying out systematic reviews to synthesise existing evidence on SMIs from RCTs; (4) compare the relative effectiveness of SMIs through NMA; (5) model the cost-effectiveness impact of SMIs; (6) analyse contextual and implementation factors and (7) develop and pilot

decision-making tools to facilitate access to and use of the most effective SMIs among key target end users.

## METHODS AND ANALYSIS

The COMPAR-EU project is divided into seven phases following to our specific objectives. Each phase is further described and shown in figure 1 (COMPAR-EU Phases and main tasks).

### Phase 1: refinement and validation of a taxonomy of SMIs

Taxonomies are formal systems for classifying multifac-eted, complex phenomena according to a set of common conceptual domains and dimensions; their use increases clarity in defining and comparing complex phenomena.[29] Several taxonomies for SMIs have been developed in the literature, but so far have focused in a specific area of the

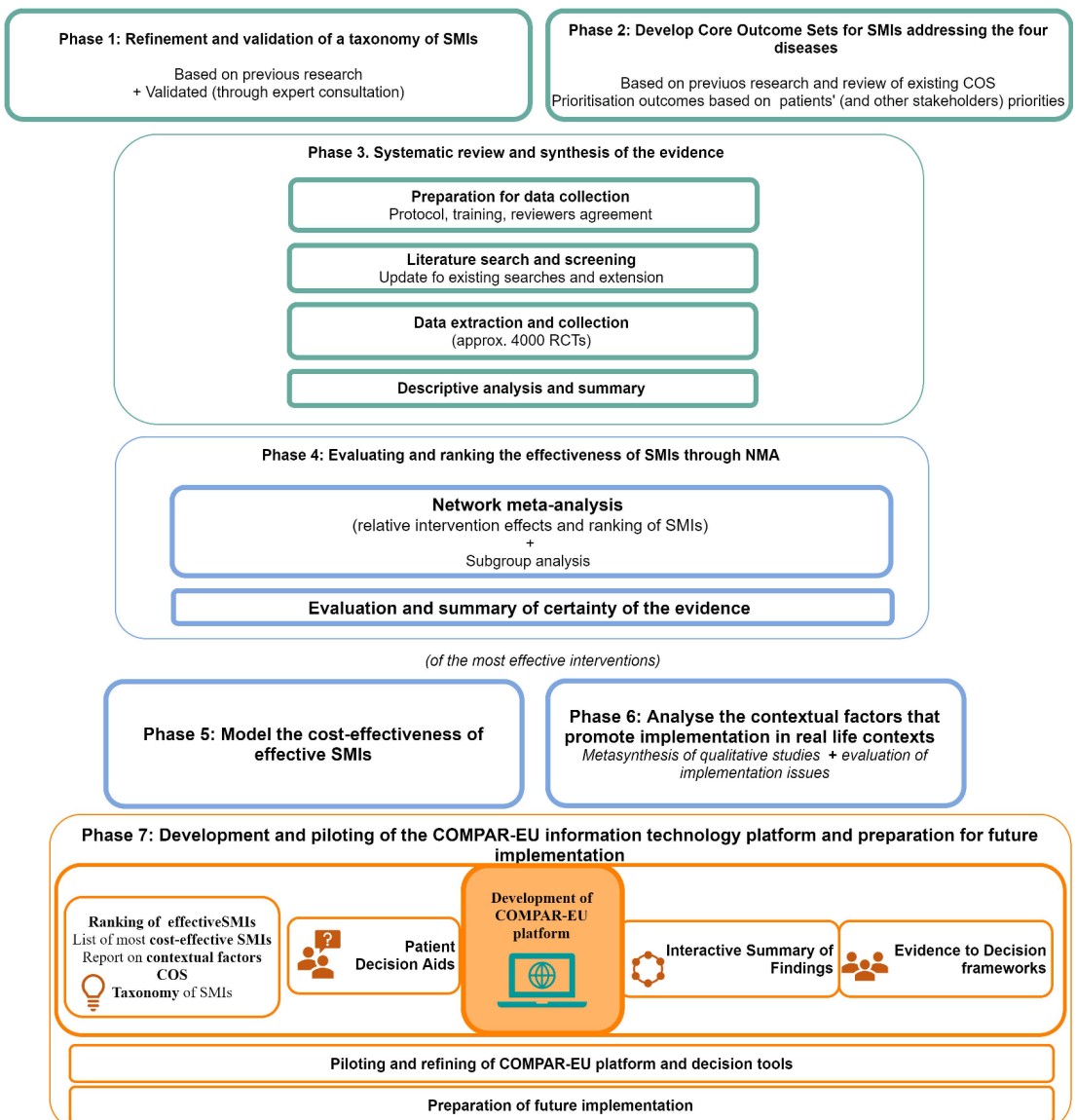

**Figure 1** *COMPAR-EU Phases and main tasks,* presents a visual summary of the main phases of the project as described in this protocol. COS, Core Outcome Set; NMA, network meta-analysis; RCTs, randomised controlled trials; SMI, self-management intervention.

intervention (eg, self-management behaviour) and have not been validated. In the first phase of COMPAR-EU, we will produce a conceptual map and classification system that will be evaluated by self-management experts and stakeholders (including patients) through a modified Delphi technique consisting of a two-round online survey. The list of candidate participants will include authors on self-management or related topics taxonomies, professional experts in self-management and patient representatives. The resulting feedback and suggestions for refinement will be integrated into a new version that will be tested by the research team when classifying the SMIs reported in the RCTs that will be included in the NMA.

### Phase 2: development of core outcome sets for each condition

Interventions can only be compared across studies when they share at least some common outcomes, and appropriate selection of outcomes is essential if research is to guide decision-making and inform policy. We will therefore develop a COS for each of the four chronic conditions included in the project.

#### Outcome catalogue

The first step is to create an exhaustive database listing outcomes reported in previous EU projects (PRO-STEP[30] and EMPATHIE[31]) and COMET, a COS database and other relevant organisations (using a snowballing technique). For each condition, outcomes will be classified into different categories, such as clinical outcomes, patient-reported outcome measures or resource utilisation.

#### Systematic review of how patients value self-management outcomes

To identify patient priorities for self-management in the selected conditions, we will conduct a scoping review, of quantitative and qualitative studies, and a series of specific overviews for each condition about what patients, and their caregivers' value on SMIs.

#### Delphi survey with patient representatives

We plan to include patient participation in our outcome selection as it has been described as a good practice to minimise the influence of power differentials between stakeholders.[32] We will use a modified Delphi survey administered to a convenience sample of patients, and patient representatives to ensure that our research addresses outcomes that matter to patients (and other stakeholders). Four panels of patients and carers, for each disease separately (5–8 members) will be given the task of prioritising relevant outcomes on a Likert scale.

#### Consensus workshop

The proposals for each of the four COSs will be presented to a panel of patients and other stakeholders (healthcare professionals, policymakers and researchers) in a workshop that aims to achieve consensus across groups on the most important outcomes to include in each COS. Participants will be previously provided with the results of the patient prioritisation process, and with the results of

the synthesis of results from the literature review. We will establish criteria to address potential discrepancies across stakeholders.

### Phase 3: systematic review—descriptive synthesis of the evidence

#### Preparations for data collection

We will develop a protocol following Cochrane guidance[33 34] and hold training sessions for those responsible for collecting data. Before the extraction process, all reviewers will be trained and undergo calibration to ensure inter-rater agreement. Additionally, in the extraction process, all data collected will be reviewed by an independent researcher to ensure quality.

#### Literature search and screening

To identify relevant RCTs, we will draw on the databases of previous European project (PRO-STEP) that identified hundreds of systematic reviews on SMIs for diabetes, obesity, COPD and heart failure. We will use these RCTs published from 2000 up to 2015 (last date of SR publication included in those projects) in our project, and update this data set through new searches in MEDLINE, CINAHL, Embase, Cochrane and PsycINFO. We will include RCTs that compared SMIs in adults with at least one of our conditions of interest (type 2 diabetes, obesity, COPD or heart failure) and are published in English or Spanish. The search will be focused in RCTs published from 2015 to 2018 to complement the findings of the SR included in PRO-STEP, with the possibility of including previous years if the systematic reviews from PRO-STEP have not covered those previous years sufficiently. An total estimate of 4000 RCTs, based on the results of a previous overview,[30] will be included for the four prioritised chronic conditions.

#### Data extraction and collection

We will extract data including patient characteristics, disease characteristics and comorbidities, intervention characteristics (guided by the taxonomy), outcomes (guided by the four COSs), results, and information on study design and risk of bias. Specific attention will be paid to subgroups of patients according to comorbidity, gender and socioeconomic variables (eg, health literacy).

#### Descriptive analysis and summary of SMIs and outcomes

SMIs and outcomes identified in the RCTs will be described and summarised to provide information on type and number of interventions, outcome results, patient characteristics and presence of comorbidities for each of the four conditions.

### Phase 4: systematic review—NMA and certainty of evidence
#### Network meta-analysis

We will initially develop theoretical models that make explicit the mechanisms through which the different SMIs operate on a given outcome. Based on our taxonomy of interventions, these models will identify hierarchies of elements (type of support provided, expected

self-management behaviour, mode of delivery of the intervention, provider, etc) that operate on the outcome to identify which components or which combinations of components are most effective.

Additionally to the standard NMA models, we will employ component NMA[35–37] to identify key intervention components and create a ranking of SMIs according to their effectiveness. During this process, it is crucial that major assumptions of NMA like transitivity (that there must be no relevant discrepancy or inconsistency between direct and indirect evidence[38]) are satisfied, as the validity of results will depend on the plausibility of the assumptions made. We will also explore the distribution of effect modifiers (eg, comorbidities, gender and socioeconomic factors) across the various comparisons. This will include differentiation, if possible, between the various forms of 'usual care' reported in the included studies.

### Evaluation and summary of the certainty of the evidence

To guide users in knowing how much confidence they can place in the summarised evidence; we will rate the certainty of evidence, obtained through the network meta-analysis, for each outcome of interest, using the Grading of Recommendations, Assessment, Development and Evaluations (GRADE) approach.[39–41] Additionally, we will apply an alternative approach, the CINeMA framework, to assess the confidence in the results by exploring how information flows in the network and how much studies at high/unclear risk of bias affect the network meta-analysis estimates. We will explore how the assessment differs between these two approaches.

### Phase 5: model the cost-effectiveness of effective SMIs

For the cost-effectiveness analysis, data on short-term effects on intermediate outcomes (eg, body mass index, HbA1C) from the NMA will serve as input for simulation models that extrapolate these effects into long-term health effects expressed in QALYs. These health benefits will be combined with costs (in 2019 Euro's) to estimate the incremental cost per QALY gained. The development of the health economic models will follow a stepped approach. First, a conceptual model of the disease will be developed, informed by a review of the literature. The conceptual model will inform which statistical techniques and models to use (eg, discrete-event models, Markov models, patient-level simulation models). These decisions will be guided by good practice guidelines from pharmacoeconomic communities of medical decision-making and the International Society for Pharmacoeconomics and Outcomes Research[42] available data and clinical expertise available within the consortium. In all four disease models, the natural course of each of the four study conditions will be modelled with incorporation of background events and disease-specific mortality and morbidity. Intervention costs will take into account societal, healthcare, and patient perspectives, and where possible, special attention will be given to societal costs that have historically been understudied (eg, productivity

gains and changes in caregiver burden and costs in life-years gained).[37] By expressing health benefits in QALYs, we will be able to produce a ranking of the most cost-effective SMIs, within and across conditions. The base cases for the models will have a societal perspective, a life-time time horizon, apply differential discounting at 4% for costs and 1.5% for health (with sensitivity analysis for equal discounting at 3%). The incremental cost-effectiveness ratios will be evaluated against the so-called v-threshold that denotes willingness to pay (WTP) for a QALY.[43] The WTP in the base case is assumed to be the median WTP of €24 226, per QALY as found in the systematic review on WTP thresholds by Ryen and Svensson,[44] with sensitivity analyses using the mean WTP threshold from that same review of € 74,159,[44] and country-specific thresholds where available. Reporting will include scenario analyses to evaluate structural uncertainty as well as univariate and probabilistic sensitivity analysis to evaluate parameter uncertainty.

### Phase 6: analyse the contextual factors that promote implementation in real life contexts

For each of the most effective SMIs identified for the four chronic conditions, we will perform a realist systematic review[45] to identify key determinants of success (or failure), such as intervention settings (eg, whether on a primary care level or hospital level care) and mechanisms (eg, engagement processes) that produce specific outcomes. At the patient level, special attention will be paid to comorbidities, gender and socioeconomic dimensions such as health literacy to better understand how these influence the implementation of the selected SMIs. We will then use a modified Delphi method, with experts on self-management and/or implementation of healthcare interventions, to establish the importance of these contextual factors in the target countries (Belgium, Germany, Greece, the Netherlands and Spain). Experts will be asked to rate the magnitude of the influence of a contextual factor of a list developed by researchers. The final list of the contextual factors will be produced in a final expert discussion.

### Phase 7: development and piloting of the COMPAR-EU information technology platform, and preparation for future implementation

We will develop an online platform that will integrate all the information and evidence synthesised during the different phases of the project. The aim is that the resources included facilitate decision-making for target end users (patients, healthcare professionals, policymakers, researchers and small and medium-sized enterprises (SMEs)). The platform will include the following GRADE-based tools:

► Evidence profiles and Interactive Summary of Findings tables[46]: these presentations will provide information in different formats about the quality of evidence, and magnitude of relative and absolute effects for each of the core outcomes identified.

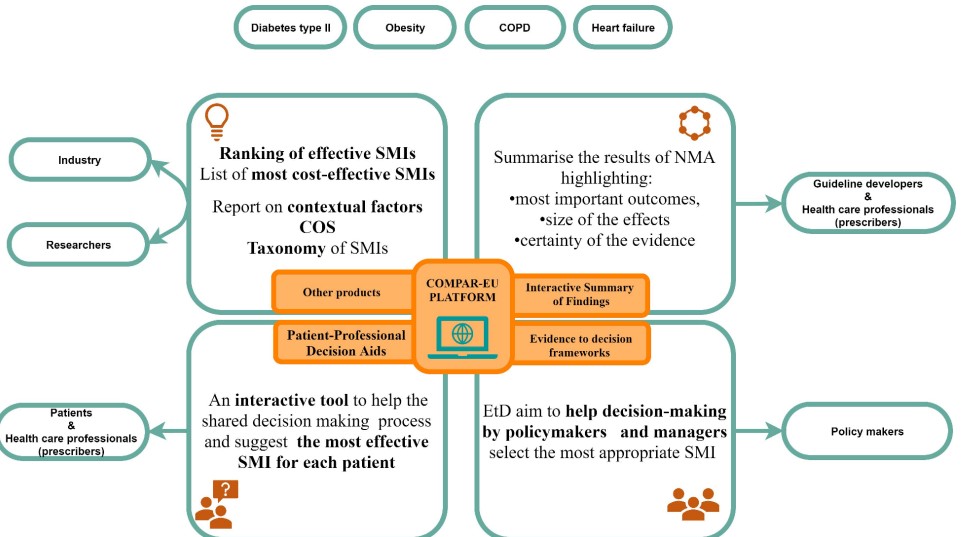

**Figure 2** COMPAR-EU platform, decision-making tools and other end products, presents a visual summary of the projects' foreseen main products and how they relate to the key stakeholders. COS, Core Outcome Set; NMA, network meta-analysis; SMI, self-management intervention.

► Evidence to Decision (EtD) frameworks: using semi-automatic templates, interactive EtD frameworks[47] will be completed for a number of priority questions that will take into account the magnitude of desirable and undesirable effects, stakeholder views on the importance of different outcomes, information on resource use and cost-effectiveness, impact on equity, and other aspects like acceptability or feasibility of the interventions. The frameworks will include draft recommendations that could be then applied or adapted to different settings.

► Patient decision aids: will be developed in plain-language for all selected situations identified in the previous phases of the study. The aids will be produced in six languages (English, French, German, Spanish, Dutch and Greek) and will include information obtained in some of the phases included in the project.

### Development of COMPAR-EU online platform

The COMPAR-EU online platform will feature structured interfaces tailored to the needs of different end users. It will be designed so that after answering a few simple questions, users will be guided to the tool or product that best suits their needs (figure 2 illustrates the main products to be integrated into the platform and expected end-users).

### Piloting and refinement of decision-making tools with different stakeholders

The COMPAR-EU platform will be piloted to gauge potential barriers to its implementation and understand how to position the decision-making tools in the target healthcare markets. Decision aids will be piloted among patients and health professionals and EtD frameworks and other products will be piloted with other stakeholders (eg, policymakers, industry and representatives of patient associations) that may be involved in implementing the tools in the five participating countries. Piloting will be organised in focus groups and user experience tests in simulated settings using techniques such as thinking-out-loud. Pilots with patient associations and clinicians will focus on the actual use of the decision aids, where pilots with industry and policymakers will focus on the potential reimbursement and integration into existing IT tools. Results of the piloting will be used to further improve the tools.

### Preparation of future implementation

Even with well-developed and user-friendly decision tools, integration in existing policy and regulatory frameworks is crucial for successful uptake of our recommendations. Using expert networks, we will assess the grey literature, current guidelines and legislation to identify and analyse relevant policy and regulatory frameworks and standards on patient participation, health technology assessment agencies, eHealth/mHealth and other topics that might influence the uptake of the self-management and decision-making tools designed within COMPAR-EU. The purpose is to ensure that current legislative, regulatory and reimbursement decisions at EU level are appropriately considered in exploiting the results of the COMPAR-EU project. We will also hold workshops with industry, pharma and SMEs to identify business opportunities resulting from the research findings that could result in long-term sustainable dissemination.

### PATIENT AND PUBLIC INVOLVEMENT

Patients are a key component of the COMPAR-EU project from start to finish and their interests are represented by the European Patient Forum (EPF), which is one of the consortium partners. Together with other stakeholders, they will be involved in different aspects of the project including prioritisation of core outcomes and testing of

interactive tools. A core group of partners, led by the EPF, will also be created to ensure co-production and establishment of explicit criteria to incorporate patient views into project outcomes, products and communication interventions. Plain-language material will also be developed to support the main end products of the project.

## ETHICS AND DISSEMINATION

The project coordinator (Avedis Donabedian Research Institute) requested the overall ethical approval for the project to our local Clinical Research Ethics Committee (CEIC) (the University Institute for Primary Care Research—IDIAP Jordi Gol). Ethical approval was granted on March 2018. Results are of interest to several stakeholder groups (patients, professionals, managers, policymakers and industry) and will be disseminated in a tailored multi-pronged approach, including the creation of an interactive platform. The data generated by the project will be managed following the Golden Open access as defined by the European Commission for Horizon 2020 research projects.[48]

## DATA SHARING STATEMENT

COMPAR-EU project will make all its anonymised data available on reasonable request, where required this will be at aggregated level. The main data dictionaries and databases generated by the project will be available on requests for uses related to research and quality improvement, and potentially for commercial exploitation, subject to approval by the consortium. Data availability and access is governed by the COMPAR-EU Data Management Plan which is aligned with the EU Open Data Initiative and the FAIR Principles.[49] Further details and information on how to access the data will be available from COMPAR-EU's project website (https://self-management.eu/).

## PATIENT AND PUBLIC INVOLVEMENT

EPF, a key umbrella organisation in patient representation and European level is a partner of this project and as such has contributed to the project from its inception and design.

Furthermore, we have planned for patient participation in key stages of the project, including the selection of outcomes for the Core Outcome Sets, advise towards patient end products and participating in the design and piloting of the COMPAR-EU platform and related decision-making tools.

### Author affiliations
[1]Avedis Donabedian Research Institute (FAD), Barcelona, Spain
[2]Universitat Autònoma de Barcelona (UAB), Barcelona, Spain
[3]Red de investigación en servicios de salud en enfermedades crónicas (REDISSEC), Barcelona, Spain
[4]Netherlands Institute for Health Services Research (NIVEL), Utrecht, The Netherlands
[5]Iberoamerican Cochrane Centre, Biomedical Research Institute Sant Pau (IIB Sant Pau), Barcelona, Spain
[6]CIBER de Epidemiología y Salud Pública (CIBERESP), Barcelona, Spain
[7]Institute for Medical Technology Assessment, Erasmus University Rotterdam, Rotterdam, The Netherlands
[8]Department of Primary Education, University of Ioannina, Ioannina, Greece
[9]Sorbone Paris Cité, Universite Paris Descartes Faculte de Medecine, Paris, Île-de-France, France
[10]OptiMedis AG, Hamburg, Germany
[11]European Patients' Forum, Brussels, Belgium

**Acknowledgements** The COMPAR-EU group for their valuable assistance in conducting this proposal.

**Collaborators** The COMPAR-EU group: Ana Isabel González (Avedis Donabedian Research Institute (FAD) and, Red de investigación en servicios de salud en enfermedades crónicas (REDISSEC), Aretj-Angeliki Veroniki (University of Ioannina, Department of Primary Education), Claudia Valli (Iberoamerican Cochrane Centre – Biomedical Research Institute Sant Pau (IIB Sant Pau)), Claudio Alfonso Rocha Calderón (Iberoamerican Cochrane Centre – Biomedical Research Institute Sant Pau (IIB Sant Pau)), Ena Niño de Guzmán (Iberoamerican Cochrane Centre – Biomedical Research Institute Sant Pau (IIB Sant Pau)),Estela Camus (Avedis Donabedian Research Institute (FAD)), Giorgos Seitidis (University of Ioannina, Department of Primary Education), Hector Pardo-Hernandez (Iberoamerican Cochrane Centre – Biomedical Research Institute Sant Pau (IIB Sant Pau) and CIBER de Epidemiología y Salud Pública (CIBERESP)), Jany Rademakers (Netherlands Institute for Health Services Research, (NIVEL)), Jessica Beltrán (Iberoamerican Cochrane Centre – Biomedical Research Institute Sant Pau (IIB Sant Pau)), Karla Salas (Iberoamerican Cochrane Centre – Biomedical Research Institute Sant Pau (IIB Sant Pau)), Kevin Pacheco-Barrios (Avedis Donabedian Research Institute (FAD)), Lyudmil Ninov (European Patients' Forum), Maria Petropoulou (University of Ioannina, Department of Primary Education), Marieke van der Gaag (Netherlands Institute for Health Services Research, (NIVEL)), Montserrat León (Iberoamerican Cochrane Centre – Biomedical Research Institute Sant Pau (IIB Sant Pau)), Nina Adrion (OptiMedis AG), Rune Poortvliet (Netherlands Institute for Health Services Research, (NIVEL)), Stella Zevgiti (University of Ioannina, Department of Primary Education), Jessica Zafra (Avedis Donabedian Research Institute (FAD)), Valentina Strammiello (European Patients' Forum).

**Contributors** MB contributed to the design of the project and prepared the manuscript of the paper. CO, MH, PA-C, MV, DM, OG, KI, CW, CC and RS contributed to the design of the project and reviewed and contributed to the manuscript of the paper.

**Funding** This work was supported by European Union's Horizon 2020 research and innovation programme under grant agreement No 754936.

**Competing interests** None declared.

**Patient consent for publication** Not required.

**Provenance and peer review** Not commissioned; externally peer reviewed.

**ORCID iD**
Marta Ballester http://orcid.org/0000-0002-7803-8354

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
