## [Reviewer comments · BMJ Open]

ARTICLE DETAILS

TITLE (PROVISIONAL)	Comparing the effectiveness and cost-effectiveness of self-management interventions in four high-priority chronic conditions in Europe (COMPAR-EU): a research protocol
AUTHORS	Ballester, Marta; Orrego, Carola; Heijmans, Monique; Alonso-Coello, Pablo; Versteegh, Matthijs; Mavridis, Dimitri; Groene, O; Immonen, Kaisa; Wagner, Cordula; Canelo-Aybar, Carlos; Sunol, Rosa

VERSION 1 – REVIEW

REVIEWER	George Patrinos University of Patras School of Health Sciences, Department of Pharmacy, Patras, Greece
REVIEW RETURNED	26-Oct-2019

GENERAL COMMENTS	This is a well-compiled description of the COMPAR-EU project
--

REVIEWER	Sue Jowett University of Birmingham, UK
REVIEW RETURNED	08-Nov-2019

GENERAL COMMENTS	Overall I am broadly happy with this protocol, but as it is such a large project with a number of different components, the individual aspects lack detailed in their proposed methodology. This may not be avoidable with word constraints. The writing is generally clear. Specific comments: P3 line 46 has an additional “a” Page 7. Are there any further details available about the approximate number of participants in the Delphi, and how are they being recruited (and where from)? P8 line 185. What type of papers will be sought for the review in Phase 2? Qualitative, quantitative or both? P9, line 218. Is there no ability for translation of relevant papers from other languages? Are many relevant papers expected to be excluded due to the language exclusions? P9 line 212. What years were the previous searches for – or can you state what years the new searches will be for? P10 This is my area of expertise and there is limited information on the economic modelling, probably due to space constraints. But details I would like to see are:  1) What currency? 2) What is the base case perspective? A number of different perspectives are mentioned. 3) What thresholds will be employed to denote cost-effectiveness? 4) Will there be expert clinical input into the model build process? 5) Will there be discounting and at what rate?
---

	6) Is sensitivity analysis planned, including probabilistic sensitivity analysis? P11 Line 264 – what is meant by this sentence “ to develop simulation models.....the first step will be to develop a conceptual model informed by a review of the literature”. How does this reference fit in with the sentence (about realist systematic reviews)? P11 line 287 – more details on this modified Delphi are needed P12 Phase 7 was the least clear and most vague to me (possible due to this being the last stage and therefore the least developed methodology) and it also needs closer proof reading. The last part on preparation or future implementation was the least clear - for example, the details of the systematic review.
--	--

VERSION 1 – AUTHOR RESPONSE

Response to editor and reviewers

We appreciate the general positive feedback received for our paper submission and the opportunity to address the issues you have raised. We include in the table below our responses to all of them, including the line/section in which to find the edit in the new version.

Editor Comments to Author:	Response	Line in paper/Section
- Please include a data sharing statement at the end of your paper, in line with BMJ Open's data policy.	Done	Line 423 onwards in marked version
- Please provide better qualities figures, ensuring the figures are not pixelated when zoomed in. We request that they have a resolution of at least 300 dpi and 90mm x 90mm of width.	Done	Uploaded
- Please provide figure legend/caption at the end of your main manuscript.	Done	Line 451 onwards in marked version
Reviewer(s)' Comments to Author: George Patrinos Institution and Country: University of Patras School of Health Sciences, Department of Pharmacy, Patras, Greece	Response	Line in paper/Section

This is a well-compiled description of the COMPAR-EU project	Thanks for the positive feedback.	
Reviewer: 2 Reviewer Name: Sue Jowett Institution and Country: University of Birmingham, UK	Response	Line in paper/Section
Overall I am broadly happy with this protocol, but as it is such a large project with a number of different components, the individual aspects lack detailed in their proposed methodology. This may not be avoidable with word constraints. The writing is generally clear. Specific comments:	Thanks for the positive feedback and remarks. We have addressed all of them, specified as follows:	
P3 line 46 has an additional “a”	Thanks for pointing this out. Corrected.	New line 49 in marked version.
Page 7. Are there any further details available about the approximate number of participants in the Delphi, and how are they being recruited (and where from)?	More detail added	New line 179 onwards in marked version.
P8 line 185. What type of papers will be sought for the review in Phase 2? Qualitative, quantitative or both?	More detail added	New line 199 in marked version.
P9, line 218. Is there no ability for translation of relevant papers from other languages? Are many relevant papers expected to be excluded due to the language exclusions?	We have limited the languages of the search due to time constraints of the project. We expect this won't cause many relevant papers to be excluded, and accept this can indeed be a limitation.	
P9 line 212. What years were the previous searches for – or can you state what years the new searches will be for?	Detail added	New line 231 in marked version.
P10 This is my area of expertise and there is limited information on the economic modelling, probably due to space constraints. But details I would like to see are:	Please see the revised section for the full revision addressing your questions	New line 282 onwards in marked version.

1) What currency?	All analyses will be expressed in 2019 Euro's.	New line 282 onwards in marked version.
2) What is the base case perspective? A number of different perspectives are mentioned.	Thank you for this request, we indeed withheld details due to word count. The base case perspective will be societal, as we feel that there has not been much attention to the productivity gains and caregiver burden (included on the cost-side by multiplying hours of care with a reference value). As many intervention are not reimbursed from the basic benefit package, we'll evaluate the patient out of pocket-costs as well when testing the assumption that the intervention is not reimbursed. We now state our base-case perspective in this section.	New line 282 onwards in marked version.
3) What thresholds will be employed to denote cost-effectiveness?	As we have a multi-country perspective, including countries without formal v- or k-thresholds, we choose to evaluate all ICERs against a v-threshold from a 2015 systematic review published in HE. We choose the median WTP as reference, with the mean WTP from that study as scenario analysis. We use the median for theoretical reasons (least affected by outliers and represent majority finding), and also because generally, self-management would be expected to face more scrutiny on the threshold than other types of interventions due to societal preferences on 'own responsibility'. We have added this section in the text: The WTP in the base case is assumed to be the median WTP of €24.226,- per QALY as found in the systematic review on WTP thresholds by Ryen & Svensson (2015), with sensitivity analyses	New line 282 onwards in marked version.

	using the mean WTP threshold from that same review of € 74.159,- and country specific thresholds where available.	
4) Will there be expert clinical input into the model build process?	The expert input will come from within the consortium, as modelers within our group have over 10 years experience in modelling COPD, Obesity, Heart failure and Diabetes. Clinical expertise is available within the advisory board who is regularly updated with the progress in the model development.	New line 282 onwards in marked version.
5) Will there be discounting and at what rate?	Yes, 4% and 1.5%, which is the standard in the Netherlands. The UK perspective of 3% discounting for both effects and costs will be used as a scenario analysis.	New line 282 onwards in marked version.
6) Is sensitivity analysis planned, including probabilistic sensitivity analysis?	Yes, PSA, CE planes and CEAC's will be presented, as well as scenario analyses and univariate DSA's.	New line 282 onwards in marked version.
P11 Line 264 – what is meant by this sentence “ to develop simulation models.....the first step will be to develop a conceptual model informed by a review of the literature”. How does this reference fit in with the sentence (about realist systematic reviews)?	Our apologies, this reference had gotten misplaced and did not belong in this section, thank you for bringing it to our attention.	New line 282 onwards in marked version.
P11 line 287 – more details on this modified Delphi are needed	More details added.	New line 325 onwards in marked version
P12 Phase 7 was the least clear and most vague to me (possible due to this being the last stage and therefore the least developed methodology) and it also needs closer proof reading. The last part on	Indeed the section has benefited from a review, that has been included in this revised version.	New line 332 onwards in marked version.

preparation or future implementation was the least clear - for example, the details of the systematic review.		
---	--	--

Other minor changes have been introduced to improve readability of the text. All of them are marked with track changes in the Main document marked version.

VERSION 2 – REVIEW

REVIEWER	Dr Sue Jowett University of Birmingham, UK
REVIEW RETURNED	18-Dec-2019
GENERAL COMMENTS	Thank you for responding to my comments, and I am happy with the revisions. Good luck with the project.